# A Long-Tailed Image Classification Method Based on Enhanced Contrastive Visual Language

**DOI:** 10.3390/s23156694

**Published:** 2023-07-26

**Authors:** Ying Song, Mengxing Li, Bo Wang

**Affiliations:** 1Beijing Key Laboratory of Internet Culture and Digital Dissemination, Beijing Information Science and Technology University, Beijing 100101, China; limengxingmx@outlook.com; 2Beijing Advanced Innovation Center for Materials Genome Engineering, Beijing Information Science and Technology University, Beijing 100101, China; 3Software Engineering College, Zhengzhou University of Light Industry, Zhengzhou 450002, China; wangb@zzuli.edu.cn

**Keywords:** long-tailed image classification, contrastive learning, data augmentation

## Abstract

To solve the problem that the common long-tailed classification method does not use the semantic features of the original label text of the image, and the difference between the classification accuracy of most classes and minority classes are large, the long-tailed image classification method based on enhanced contrast visual language trains the head class and tail class samples separately, uses text image to pre-train the information, and uses the enhanced momentum contrastive loss function and RandAugment enhancement to improve the learning of tail class samples. On the ImageNet-LT long-tailed dataset, the enhanced contrasting visual language-based long-tailed image classification method has improved all class accuracy, tail class accuracy, middle class accuracy, and the F_1_ value by 3.4%, 7.6%, 3.5%, and 11.2%, respectively, compared to the BALLAD method. The difference in accuracy between the head class and tail class is reduced by 1.6% compared to the BALLAD method. The results of three comparative experiments indicate that the long-tailed image classification method based on enhanced contrastive visual language has improved the performance of tail classes and reduced the accuracy difference between the majority and minority classes.

## 1. Introduction

Image classification [1] is the earliest application of machine learning in the field of computer vision, and is the foundation of other visual tasks such as object detection and instance segmentation. Due to the rich semantic information contained in images (such as multiple targets, scenes, and behaviors), the characteristics closest to human perception and expression ability, and the gradual optimization of the performance and cost of visual sensors (mainly cameras), image classification and its derived detection, segmentation and other visual algorithms are gradually being applied in fields such as healthcare, transportation, and signal processing [2]. However, in the application process, due to the unique nature of the actual environment, some difficult to solve problems are gradually encountered.

In image classification tasks, input data are manually collected and annotated, and through human intervention, the amount of data in each category are balanced as much as possible, with no significant difference in sample size among different categories. The manually balanced dataset simplifies the requirements for algorithm robustness; but with the gradual increase in the focus categories, maintaining the balance among various categories will bring exponential growth in acquisition costs. For example, if an animal classification dataset is to be built, it is easier to collect millions of pictures from common data such as cats and dogs. However, considering the balance of the dataset, it is also necessary to collect the same amount of samples for rare animals such as snow leopards. With the increase in the rarity of the category, the collection volume tends to show exponential growth, as shown in Figure 1.

In practical applications, such as facial recognition, species classification, autonomous driving, medical diagnosis, drone detection, and other fields, there is a problem of long-tailed class imbalance [3]. For example, for autonomous driving, the data on normal driving will account for the vast majority, while there are very little data on actual abnormal situations/car accident risks. For medical diagnosis, the number of people with specific diseases is also extremely uneven compared to the normal population. However, this type of imbalance problem often makes the training of deep neural networks very difficult. Classification and recognition systems that directly use long-tailed distribution data for training often tend to lean towards the head class data, making them insensitive to tail class features during prediction and affecting the correct judgment of the system [3]. In traditional methods, a series of common methods to mitigate performance degradation caused by long-tailed distribution data are based on categories re-balancing strategy, including re-sampling training data and re-weighting to redesign the loss function [3]. These methods can effectively reduce the bias of the model to the head class in the training process, thus producing a more accurate classification decision boundary. However, because the distribution of the original data is unbalanced, and the over parameterized deep networks are easy to fit to this composite distribution, they often face the risk of tail class overfitting and head class under-fitting.

Given that the problem of class imbalance in long-tailed distribution datasets is very widespread in practical tasks, it is crucial to train high-performance network models from many images that follow the long-tailed distribution. Moreover, the difference in class distribution between training and testing data will greatly limit the practical application of neural networks. This research topic has important practical significance and is an important paradigm for promoting the implementation of deep neural networks in model implementation. How to effectively utilize long-tailed data to train a balanced classifier is a key issue. From a practical implementation perspective, this study will improve the speed of data collection and reduce collection costs. This article explores effective contrastive learning strategies to learn better image representations from imbalanced data, in order to better apply them to long-tailed image classification. We hope to provide better development ideas for the application of image classification in today’s gradually developing image technology. The ECVL method proposed in this article utilizes text label information of images for pre-training to assist in image classification, transforming image recognition problems into visual language matching problems. After pre-training, model training is conducted separately for the head class and tail class, which can improve the performance of the tail class without sacrificing the performance of the head class.

## 2. Related Work

The research contents of this paper is long-tailed distribution image classification. Long-tailed image classification methods mainly include data re-sampling, data re-weighting, data augmentation, transfer learning, and ensemble learning. Table 1 shows the advantages and disadvantages of mainstream long-tailed classification methods.

### 2.1. Data Re-Sampling

Data re-sampling solves the problem of long-tailed distribution image classification from the data level. Re-sampling is the most widely used method [4,5] in processing long-tailed distribution image classification in depth learning, mainly including over-sampling [6,7], under-sampling [8,9,10] and mixed sampling [11,12].

The oversampling method mainly reduces the imbalance between the head class and the tail class by increasing the number of samples of the tail class [6,7]. Inspired by this, Gupta et al., proposed the repeated factor sampling method [13], which performs a re-balancing operation on the training data by increasing the sampling frequency of the tail image. Peng et al., proposed the soft box sampling method [14], which utilizes class perception sampling to calculate the replication factor for each image based on the distribution of labels, and replicate the images according to the specified number of times to solve the problem of class imbalance. Hu et al., designed an instance level class balancing scheme [15] to balance instance level samples of original images. The balanced samples are learned to use meta-modules, transferring knowledge from the data-rich header to the data poor tail. In 2020, Wu et al., proposed non maximum suppression re-sampling [16], which adaptive adjusts the threshold for non-maximum suppression based on the label frequencies of different classes, in order to retain more candidate target classes from the tail class and balance the data distribution by suppressing candidate target classes from the head class. According to the principle of the oversampling method, this method simply repeats the positive example, which will cause overemphasis on the positive example, and it is easy to over fit the positive example. In 2022, Park et al., proposed a oversampling method based on feature dictionary [6], and built a feature dictionary through a pre-trained feature extractor. The method of synthesizing samples based on feature dictionaries enriches the diversity of minority class data by fine-tuning classifiers. In 2023, Li et al., proposed some oversampling based on subspaces [7]. This method believes that each type of sample is formed by common and unique features, and these features can be extracted through subspace. In order to obtain balanced data, map images belonging to minority categories are oversampling to more accurately describe minority categories. Balanced data are obtained by restoring the generated subspace product to the original space.

Compared with the oversampling method, the under-sampling method reduces the imbalance between the head class and the tail class by reducing the number of samples of the head class [8,9,10]. In 2020, the bilateral branch network (BBN) [17] developed conventional learning branches and re-balanced branches, using a new bilateral sampling strategy to address class imbalance issues. Uniform sampling was applied to simulate the original long-tailed training for conventional branches, and a reverse sampler was applied to sample more tail class samples for re-balanced branches to improve tail class performance. In 2021, Lee et al., proposed a framework for classifying unbalanced data using undersampling and convolutional neural network [8], created a balanced training set through under-sampling, and then used convolutional neural network for training. In 2021, Zang et al., proposed a feature and sampling adaptive strategy [18], which used model classification loss to adjust the sampling rates of different classes on the validation set, thereby sampling more tail class samples with insufficient representation. In 2022, Lehmann et al., proposed a subclass based under-sampling method [9], which selects samples from all subclasses of a class for under-sampling, and identifies subclasses by clustering the advanced features of the CNN model. In 2023, Farshidvard et al., proposed a method based on under-sampling and ensemble [10], which divides most classes into clusters, so that there are no minority class samples in the majority samples of each cluster, while controlling the size of each cluster.

Mixed sampling [11,12] is a method of combining oversampling and under-sampling to achieve sample balance. In 2020, Ding et al., proposed a KA integration method of under-sampling and oversampling [11], under-sampling the majority of classes through the kernel based adaptive synthesis method, and oversampling the minority classes at the same time, generating a set of balanced datasets to train the corresponding classifiers separately, and the final results will be voted by all these trained classifiers. By combining under-sampling with oversampling in this way, KA ensemble is good at solving the class imbalance problem with large imbalance rate [11]. In 2022, EF Swana et al., studied the use of a Naïve Bayes classifier, support vector machine, and k-nearest neighbors together with synthetic minority oversampling technique, under-sampling, and the combination of these two re-sampling techniques for fault classification with simulation and experimental imbalanced data.

Re-sampling is the most common method to solve the problem of long-tailed distribution image classification. However, these classic methods generally have poor results. For example, in the case of Oversampling of tail classes, it may lead to overfitting of tail classes [6,7], and if there are errors or noises in the samples of tail classes, oversampling may exacerbate these problems. Under-sampling may lead to insufficient learning of head classes [8,9,10] and may result in the loss of valuable data in the head classes. For extremely imbalanced long-tailed data, under-sampling methods often lose a large amount of information due to the significant difference in data volume between the head class and tail class.

### 2.2. Data Re-Weighting

The data re-weighting strategy aims to minimize the total cost of the classifier, and solves the problem of imbalanced data classification by adjusting the loss values of different classes during training and increasing the attention of minority class samples during model learning.

In 2017, Lin et al., proposed focal loss [19]. During training, this loss function can automatically reduce the weight of the head class, making the model focus on learning the tail class. In 2017, Hermans et al., proposed the triplet loss function [20], and used the Gradient descent to train samples with small differences. In 2019, class balanced loss (CB) [21] introduced the concept of effective sample size, which alleviated the problem of class imbalance by forcing a class balance re-weighting term that is inversely proportional to the number of effective samples in the class. In 2019, Cao et al., proposed label-distribution-aware margin loss (LDAM) [22], which involves the model learning the initial feature representation before re-weighting. In 2020, the distribution balance loss [23] was alleviated by a new tolerant regularization method to alleviate gradient over suppression. At the same time, it also evaluates the difference between the expected sampling frequency and the actual sampling frequency for each class, and then uses the quotient of these two frequencies to recalculate the weighted loss values for different classes. In 2020, equalization loss [24] proposed to directly reduce the weight of loss values for tail class samples when the tail class samples are negative sample pairs for a large number of head class samples. In 2021, equalization loss v2 [25] extended equalization loss by introducing a new gradient re-weighting mechanism that dynamically increases the weight of positive gradients and decreases the weight of negative gradients for model training on each subtask. Seesaw loss [26] re-balances the positive and negative gradients of each class using mitigation and compensation factors. LADE [27] introduces label distribution decoupling loss to disentangle the learning model from the long-tailed training distribution, and then adapts the model to any test class distribution when the test label frequency is available.

Balanced meta-softmax [28], optimizing sample distribution by adjusting the model on the validation set. The progressive margin loss function [29] uses two margin items to adjust the classification margin of long-tailed learning. Sequential margins extract discrimination features and maintain class order relationships. The variational margin gradually suppresses the head class and handles class imbalance in long-tailed training samples. The adversarial robust long-tailed classification method [30] re-balances data through a scale invariant classifier and boundary adjustments during the inference process.

Although the data re-weighting method alleviates the imbalance in gradient proportion caused by long-tailed distribution, for some extreme cases, such as when the sample proportion of tail categories is very small, the recognition accuracy of tail classes is still at a low level.

### 2.3. Data Augmentation

Data augmentation aims to utilize a series of data augmentation techniques to enhance the size and quality of the dataset [31,32] for model training. In 2021, Zhang et al., proposed a data augmentation method based on neighborhood risk minimization [33], which helps correct overconfidence in the model. In the decoupling training scheme, this method has a positive impact on representation learning and a negative or negligible impact on classification learning. Based on these observations, data mixup is used in the decoupling training scheme to enhance representation learning.

In addition, the Remix method also adopts a method for long-tailed learning and introduces a re-balancing hybrid enhancement method to enhance tail classes. In 2021, Li et al., proposed Meta semantic augmentation (MetaSAug) based on meta learning [34], using a variant of implicit semantic data augmentation (ISDA) [35] to enhance tail classes. ISDA obtains semantic direction by estimating the covariance matrix of sample features, and translates deep features along multiple semantic meaningful directions to generate a diverse enhancement samples. However, due to insufficient tail class samples, it is impossible to estimate the covariance matrix of tail class. To solve this problem, MetaSAug explores meta learning to guide the learning of each class’s covariance matrix. In this way, the covariance matrix of tails can be estimated more accurately, thus generating rich tail class feature information. Although data augmentation methods enhance the diversity of training samples, they are prone to introducing noise and ambiguity during the training process.

### 2.4. Transfer Learning

Transfer learning is to transfer knowledge from the source domain to enhance model training in the target domain. The source domain is a different domain from the test sample, but it has rich supervisory information. The target domain is the domain where the test sample is located, with no labels or only a few labels. There are mainly four transfer learning schemes in the deep learning processing of long-tailed distribution image classification, namely, head-to-tail knowledge transfer, model pre-training, knowledge distillation and self-training.

The knowledge transfers from beginning to end is to transfer the knowledge of the head class to the tail class to enhance model performance. Yin et al., proposed feature transfer learning (FTL) [36], which uses the intra-class variance knowledge of the head class to enhance the characteristics of the tail class samples, so that the tail classes features have higher intra-class variance, so that the tail class gets better performance. The LEAP [37] method proposed by Liu et al., constructs a “feature cloud” for each class by adding tail class samples with certain interference in the feature space, seeking to transfer the knowledge of the head class feature cloud to enhance the intra-class variation in the tail class feature cloud. This method effectively alleviates the problem of inter class feature variance distortion. The online feature enhancement method [38] uses class activation mapping to decouple sample features into specific class features and uncertain class features, and combines the class specific features of the tail class samples with the class unknown features of the head class samples to enhance the tail class. Then, using all enhanced and original features, the model classifier is fine-tuned using a re-balancing sampler to achieve better long-tailed learning performance.

Model pre-training is one of the commonly used methods for deep learning model training. Domain-specific transfer learning [39] uses all long-tailed samples to pre-train the model, and then fine tune the model on the training subset of classes’ balances. Slowly transfer the learned features to the tail class to achieve a more balanced performance among all classes. In addition, the self-supervised pre-training method [40] first uses self-supervised learning (such as comparative learning [41] or rotation prediction [42]) for model pre-training, and then carries out standard training on long-tailed data. This scheme is also used to process long-tailed data with noise labels [43]. The proposal of the visual and language pre-training dataset (Conceptual 12 M [44]) has promoted the development of visual language models in the field of long-tailed recognition.

Knowledge distillation is the training of student models using the output of well-trained teacher models. The learning from multiple experts (LFME) method [45] divides the entire long-tailed distribution dataset into several subsets with less imbalanced classes, and train multiple experts with different sample subsets. Based on these experts, the LFME method utilizes adaptive knowledge distillation methods and selects difficult course examples to train a unified student model. The routing diversity distribution-aware experts (RIDE) [46] introduces a knowledge distillation method on the basis of a multi-expert framework to reduce the parameters of the multi-expert model by learning a student network model with fewer experts. The self-supervised distillation method [42] has invented a new self-distillation scheme to enhance decoupling training. The decoupling training scheme trains a calibration model based on supervised and self-supervised information, and then uses the calibration model to generate soft labels for all samples. Afterwards, a new student model is extracted using the generated soft labels and the original long-tailed labels, and finally a new classifier fine-tuning stage is entered. In addition, the distillation virtual instance method [47] uses a class equilibrium model as the teacher model to solve the long-tailed classification problem.

The purpose of self-training is to learn well-performing models from some labeled samples and numerous unlabeled samples. The Class balancing self-training (CReST) method [48] studied self-training in long-tailed classification and found that the supervised model has high classification accuracy for tail classes. Based on this discovery, CReST proposes to select more tail class samples for online pseudo labeling in each iteration, enabling the retrained model to achieve better performance on tail classes. The MosaicOS method [49] pre-trains the model using scene centered images labeled in the original detection dataset. The pre-trained model is fine-tuned in two stages: first, the pseudo labeled object centered image is fine-tuned, and then the original labeled scene centered image is fine-tuned, which can alleviate the negative impact of data differences and effectively improve long-tailed learning performance.

Due to the introduction of additional knowledge, the transfer learning method improves the performance of the tail class without sacrificing the performance of the head class, but the performance improvement is not obvious when the difference between the head class and the tail class is large. The lack of sufficient tail class samples is one of the key problems of long-tail learning, and the related methods of transfer learning to deserve further exploration.

### 2.5. Ensemble Learning

The method based on ensemble learning solves the learning problem of long-tailed distribution image by strategically generating and combining multiple network modules (multiple experts). Long-tailed multi-label visual recognition method [50] explored a bilateral branch network solution to long-tailed multi-label classification, used the sigmoid cross-entropy loss function to train each branch for multi-label classification, and forced the use of logit consistency loss to improve the consistency of the two branches.

The all complete experts (ACE) method [51] divides all classes into several subsets: one subset contains all classes, one contains intermediate and tail classes, and the other only has tail classes. ACE train multiple experts with different class subsets, and uses distributed adaptive optimizers to adjust the learning rate of different experts. In 2022, the ResLT [52] method proposed by Cui et al., also had an idea similar to ACE. The Test time aggregating diverse experts (TADE) [53] explores multiple expert schemes to handle long-tailed recognition problems, where the distribution of test classes can be uniform or long-tailed. TADE provides two solutions: one is a diverse expert learning strategy that can train experts with different class distributions based on the characteristics of long-tailed distribution datasets. The second is the testing time expert aggregation strategy, which can use self-supervised methods to aggregate multiple experts to process data of various unknown test distributions. The methods based on ensemble learning usually achieve better performance on the head and tail classes. However, such methods often result in higher computational costs due to the use of multiple experts.

## 3. Method

Real data often follow a long-tailed distribution, with the head class dominating the training and the tail class having only a few samples, which is a major challenge in the field of image classification. The existing methods either use manually balanced datasets (such as ImageNet) or develop more robust algorithms to process data, such as class re-balancing strategies and network module improvements.

Although the above methods are effective for long-tailed distribution datasets, they sacrifice the performance of the head class at different levels. To address these limitations, researchers have turned to exploring new network architecture training paradigms. Long-tailed classification models typically include two key parts: feature extractors and classifiers. For each component, there are corresponding methods, either designing better classifiers [37,54], or learning reliable representations [55,56]. In terms of the new training framework, existing work attempts to divide one stage of training into two stages. For example, the learning process of decoupling training method [57] is decoupled into representation learning and classifier training. In addition, the ensemble learning scheme [51,53] first learns multiple experts with different data subsets, and then combines them to deal with the long-tailed distribution image classification problems. However, these methods all use a limited set of predefined labels to train the model, ignoring the availability of semantic feature information in the original label text of the image. After research, it was found that previous work was most limits to a predetermined approach when dealing with imbalanced datasets, which relied entirely on visual models and completely ignored the semantic features of the original label text rich in the image itself. This may be a promising solution to impose additional supervision on insufficient data sources.

The large-scale visual-language pre-training model provides a new approach for image classification. Through open vocabulary supervision, pre-trained visual-language models can learn powerful multi-modal representations (input information can be expressed in multiple ways). Utilize semantic similarity between visual input and text input to transform visual recognition into a visual-language matching problem. Comparative visual language models such as CLIP [58] and ALIGN [59] provide new ideas for long-tailed classification tasks. The feature extractors of these models integrate image and text modalities, focusing on learning feature matching between different modalities. They have strong robustness, but lack the ability to model complex interactions between images and text.

Due to the significant difference in classification accuracy between majority and minority classes in commonly used long-tailed classification algorithms, the failure to utilize the semantic features of the original image label text, and the inability of existing contrastive visual-language models to model complex interactions between images and text, this paper proposes an enhanced contrastive visual language long-tailed image classification algorithm (ECVL). The algorithm uses a two-stage training method, designs the loss function for text and image retrieval, respectively, uses enhanced momentum to compare the loss function to measure the learning degree of samples, and applies random enhancement to the categories with insufficient learning degree to further strengthen the learning of the model for minority samples.

### 3.1. The Overall Framework

Similar to common contrastive visual-language models, the ECVL long-tailed image classification algorithm uses a two-stage training approach to transform visual recognition into a visual-language matching problem through similarity between visual and text inputs. The first stage mainly uses the visual features of the image and the semantic features of the original label text to train for most categories. The second stage first uses class balance for a few categories, and then uses linear adapters to carry out differentiated training. Finally, use the enhancement momentum to compare the loss function to measure the memory of the model for samples. For samples with insufficient memory, use the RandAugment [60] to select random enhancement methods. Enhancing breadth can further enrich feature representation.

### 3.2. Contrasting Visual-Language Pre-Training Model

Compare visual language models with a dual encoder architecture, including a language encoder $L_{enc}$ and a visual encoder $V_{enc}$. Given an input image I, use $V_{enc}$ extracts the visual features of image I using the equation shown in (1). Similarly, use $L_{enc}$ encodes the input text sequence T as its corresponding text feature, as shown in the Equation (2).
(1)$$f_{v}=V_{enc}(I)\in R^{d_{v}}$$
(2)$$f_{l}=L_{enc}(T)\in R^{d_{l}}$$

After extracting the features of each modality, use two transformation matrices $W_{v}\in R^{d_{v}\times d}$ and $W_{l}\in R^{d_{1}\times d}$ project the original visual and textual features into a shared embedding space, where *v* and *u* are d-dimensional normalized vectors, as shown in Equation (3).
(3)$$v=\frac{W_{v}^{⊤}f_{v}}{∥W_{v}^{⊤}f_{v}∥},u=\frac{W_{l}^{⊤}f_{l}}{∥W_{l}^{⊤}f_{l}∥}$$

In the pre-training stage, for text–image pairs in a batch, the training goal is to shorten the distance between the same category and different categories, $L_{v\to l}$ for text retrieval, $L_{l\to v}$ for image retrieval, where τ indicates that the temperature exceeds the parameter, and τ represents the number of text–image pairs in a batch. $L_{v\to l}$ and $L_{l\to v}$ are as shown in Equations (4) and (5).
(4)$$L_{v\to l}=-\frac{1}{N}\sum_{i}^{N}log\frac{\exp\left(v_{i}^{⊤}u_{i}/\tau \right)}{\sum_{j=1}^{N}exp(v_{i}^{⊤}u_{j}/\tau )}$$
(5)$$L_{l\to v}=-\frac{1}{N}\sum_{i}^{N}log\frac{\exp\left(u_{i}^{⊤}v_{i}/\tau \right)}{\sum_{j=1}^{N}exp(u_{i}^{⊤}v_{j}/\tau )}$$

By converting the category labels of an image into a text sequence of “A photo of a {Class}”, the matching score between the target image and the text sequence of all categories can be obtained. The category with the highest score is selected as the final predicted category. The normalized test image features are represented as v, and the normalized text features are represented as $\left\{u_{1},\cdots ,u_{K}\right\}$. Therefore, the category probability of the test image is shown in Equations (5) and (6), where $p_{i}$ represents the probability of class i, and K represents the total number of candidate classes. Finally, the text label with the highest probability will be selected as the prediction result.
(6)$$p_{i}=\frac{exp(v^{⊤}u_{i})/\tau }{\sum_{j=1}^{K}exp(v^{⊤}u_{j})/\tau }$$

### 3.3. Balanced Linear Adapters

The performance of contrastive visual-language models on the head and tail classes is balanced, while traditional contrastive learning methods such as PaCo [61] have lower performance on the tail classes due to a lack of training samples. Inspired by the zero-shot classification ability of visual-language comparison models, improvements were made on the basis of CLIP. The training of long-tailed data is divided into two stages. The first stage fully utilizes existing training data and ensures the performance of most categories, while the second stage focuses on improving the learning ability of a few categories. These two stages aim at the long-tailed and balance training samples, respectively, and refine the comparison loss function.

According to the research results proposed by Gururangan [62] et al., in Phase I, model pre-training with domain adaptation and task adaptation can greatly improve the performance of the target NLP task. Similarly, this applies equally to image classification tasks. In stage one, pre-training using the contrastive visual-language backbone model on the long-tailed target dataset is also beneficial for learning most class samples, making full use of available training data. Since the input of the model in Phase I is to process image category labels into text sequences, the contrastive loss function used in the pre-training is Equation (4). The parameters of the text encoder and image encoder are updated instantly during training. After stage one training, most classes usually achieve good results, while minority class samples require stage two balance training. The processing of the stage model is shown in Figure 2.

Due to the insufficient sample size and limited data for the tail category, direct training on the backbone in Phrase II will result in overfitting. Therefore, in this stage, pre-training is not conducted on the backbone, but instead, linear adapters and enhancements are used to optimize the visual language representation of a few category samples for momentum contrast loss. As shown in Figure 3, the processing of the semantic features of the original label text is the same as that of Stage I. Assuming the original image feature is f, the weight matrix of the linear adapter is $W\in R^{d\times d}$. The offset is $R^{d}$, and the processed image features can be expressed as Equation (7).
(7)$$f^{⋆}=\lambda \cdot ReLU(W^{⊤}f+b)L_{DCVL}+(1-\lambda )f$$
where λ, the residual factor, is used to dynamically combine the image features after fine-tuning in the second stage with the original image features in the first stage.

The enhanced momentum contrastive loss function is used to measure the learning of the model for samples. Assuming $x_{i}$ is the training sample on the long-tailed dataset, $x_{i}$ The comparison loss is expressed as $L_{i}$. $\left\{L_{i,0},\ldots ,L_{i,t},\ldots ,L_{i,T}\right\}$ represents the tracking loss value $L_{i}$ among T Epochs. Based on this, define the moving average momentum loss, as shown in Equation (8).
(8)$$L_{i,0}^{m}=L_{i,0},L_{i,t}^{m}=\beta L_{i,t-1}^{m}+(1-\beta )L_{i,t}$$

The β is a hyperparameter that represents the smoothness of the loss. After training T Epochs using the above moving average momentum loss, the set of momentum losses for each sample can be obtained as $\left\{L_{0,t}^{m},\ldots ,L_{i,t}^{m},\ldots ,L_{N,t}^{m}\right\}$, where *N* is the number of training samples in the dataset. Finally, the definition of momentum loss is normalized as follows, as shown in Equation (9):(9)$$M_{i,t}=\frac{1}{2}\left(\frac{L_{i,t}^{m}-{\overset{-}{L_{t}}}^{m}}{\max{\left\{|L_{i,t}^{m}-{\overset{-}{L_{t}}}^{m}|\right\}}_{i=0,\ldots ,N}}+1\right)$$
where ${\overset{-}{L_{t}}}^{m}$ represents the average momentum loss of the t Epoch. The range of $M_{i}$ normalized values is $\left[0,1\right]$, with an average value of 0.5, reflecting the model’s level of sample memory. To promote model learning, use $M_{i}$ to control the occurrence and intensity of enhancement indicators. The specific approach follows RandAugment [61], randomly selecting k types of enhancements and using probability $M_{i}$ and intensity $\left[0,{\;M}_{i}\right]$ apply each enhancement. Assuming that the enhancement set defined by RandAugment is $A=\left(A_{1},\ldots ,A_{j},\ldots ,A_{K}\right)$, where K is the enhancement amount, *k* enhancements are applied in each step. On this basis, define a memory enhancement function, as shown in Equation (10).
(10)$$\begin{array}{c} \Psi (x_{i};A,M_{i})=a_{1}(x_{i})\ldots a_{k}(x_{i}),\\ a_{j}(x_{i})=\left\{\begin{array}{c} A_{j}\left(x_{i};M_{i}\zeta \right)\\ x_{i} \end{array}\right.\begin{array}{cc} \; & u∼U(0,1)\&u<M_{i}\\ & \mathrm{other} \end{array} \end{array}$$
where ζ sampling is from uniformly distributed $U\left(0,1\right)$. $A_{j}\left(x_{i};M_{i}\zeta \right)$ represent $x_{i}$ undergoes the *j* enhancement with a strength of $M_{i}\zeta$. Apply the selected k enhancements in sequence in A. For simplicity, use $\Psi \left(x_{i}\right)$ to represent $\Psi (x_{i};A,M_{i})$. In this paper, the enhanced momentum loss function is shown in Equation (11).
(11)$$L_{DCVL}=\frac{1}{N}\sum_{i}^{N}-log\frac{\exp\left(\frac{f(\Psi (x_{i}){)}^{⊤}f(\Psi (x_{i}^{+}))}{\tau }\right)}{\Sigma_{x_{i}^{\prime }\in X^{\prime }}exp(\frac{f(\Psi (x_{i}){)}^{⊤}f(\Psi (x_{i}^{\prime }))}{\tau })}$$
where $X^{\prime }$ represents $X^{-}\cup \left\{x_{i}^{+}\right\}$, $x_{i}$ and $x_{i}^{+}$ represents two views of a sample, $x_{i}^{\prime }\in X^{-}$ is a view of other samples. Intuitively, the enhanced momentum contrastive Loss function is used to measure the memory of the model for the samples, and adaptively allocate appropriate enhanced strength for the samples with insufficient memory.

In the training process of stage 2, to avoid the model deviating from the head class, a class balance sampling strategy [8] is still used to construct a balanced training sample set. Assuming there are K classes in the target dataset to form a total of N training samples. The number of training samples for class j is expressed as $n_{j}$. Then, use Equation (12) to represent N.
(12)$$N=\sum_{j=1}^{K}n_{j}$$

Assuming that classes are sorted in descending order, the long-tailed distribution means $n_{i}\geq n_{j}$ (i < j and $n_{1}\gg n_{K}$). For class balanced sampling, the probability of sampling each data point from class j is defined as $q_{j}=1/K$. In other words, to construct a balanced training sample set, first select a class from K candidate objects, and then sample a data point from the selected class. Finally, through stage two, use $L_{v\to l}$ to fine tune the balanced training data.

### 3.4. Algorithm Description

Based on the introduction of the ECVL long-tailed image classification algorithm in the previous text, this section mainly introduces the training process of the long-tailed image classification algorithm based on enhanced contrastive visual language in two different stages: stage one and stage two, as shown in Algorithms 1 and 2.

Algorithm 1 is the training process for model stage one, which simultaneously trains the visual and language branches of the visual language model. In each Epoch, it is preferred to input images and corresponding category text information; Afterwards, the visual features of the image and the semantic features of the original label text are extracted using Equations (1) and (2), respectively. Additionally, then use $L_{v\to l}$ to perform text retrieval using $L_{l\to v}$ to perform image retrieval to obtain associated image and text information. Finally, use Equation (6) to predict the image category, and evaluate the prediction results using evaluation indicators after the classification is completed.
**Algorithm 1: Phrase I**input: *I_input_* = {*images, labels*}, *T_input_* = {*texts, labels*}output: *model_weight_* 1: **for**
*epoch = 1 to max_epoch*
**do** 2:  *T = Encode(labels, text)* 3:  *I = Encode(labels, images)* 4:   *train(model, I)* 5:   *Eval(model, images, labels)* 6:   *Logits(I, T)* 7:   *pth_epoch_ = {weight}* 8: **end for**

Algorithm 2 is the training process for model stage 2. The model first balances a few types of samples, and then fine tunes the linear adapter. After fine tuning, it uses the enhanced momentum loss function described according to Equation (11) to evaluate the sample learning situation. For samples with insufficient representation of learning features, RA random enhayncement is used. Finally, the features learned in these two stages are dynamically fused and output.
**Algorithm 2: Phrase II**input: *I_input_* = {*images, labels*}, *T_input_* = {*texts, labels*}, *model_stage1_*output: *weight* 1: *model = load(best_model)* 2: **for**
*epoch = 1 to max_epoch*
**do** 3:  **if**
*epoch >= 2*
**then** 4:     *I = Rebalance(Momentum)* 5:  **end if** 6:  *Momentum = model(I, labels, epoch)* 7:  *train(model, I)* 8:  *eval(model, images, labels)* 9:  *Logit(model, I, T)* 10:   *pth_epoch_ = {weight}* 11: **end for**

## 4. Experiments

The ECVL algorithm takes 229 s to infer 100 images on a single NVIDIA A100 40 G GPU. In order to verify the performance of the proposed ECVL long-tailed image classification algorithm, experiments were carried out on three common long-tailed distribution datasets CIFAR100-LT, Places-LT and ImageNet-LT to analyze the performance of this algorithm, and ablation experiments were conducted to prove the role of enhanced momentum in comparison with the loss function and random enhancement. This section may be divided by subheadings. It should provide a concise and precise description of the experimental results, their interpretation, as well as the experimental conclusions that can be drawn.

### 4.1. Long-Tailed Image Datasets

The dataset used in the comparative experiment of this paper is a common dataset in the field of long-tailed image classification, including CIFAR100-LT [27], Places-LT [63], and ImageNet-LT [63]. The composition of each dataset is introduced below.

#### 4.1.1. CIFAR100-LT

CIFAR100-LT [27] is the dataset obtained by long-tailed of the dataset CIFAR100. It is created by reducing the number of training samples of each class through the exponential function, and the test set remains unchanged.

#### 4.1.2. Places-LT

Places-LT [63] is a dataset obtained by long-tailed transformation based on the Places [55] dataset. The Places dataset contains 10 million images classified by scene, and the label of the sample represents the meaning of the scene. It is currently the largest scene dataset in the world with the largest sample size, as shown in Figure 2. The long-tailed rate of the training set in the Places-LT dataset is 996, and the number of categories is 365. The total sample size in the training set is 62,500, and the sample size in the test set is 7300. The category with the largest sample size in the training set is 4980, while the category with the smallest sample size is 5. The ratio of the maximum to minimum sample size is 996, making it the dataset with the largest long-tailed rate used in this article.

#### 4.1.3. ImageNet-LT

ImageNet-LT [63] was obtained through the long-tailed ImageNet dataset, with a total of 1000 categories. The total number of samples in the dataset exceeds 186 K, with 116 K training samples, 20 K validation samples, and 50 K testing samples. In ImageNet-LT, the long-tailed rate in the training set is 256, the maximum class sample size is 1280, and the minimum class sample size is 5. This dataset simulates the distribution of long-tailed data commonly found in real life. The data in the training set are divided into three parts. The header category contains categories with a sample size greater than 100, the middle category contains categories with a sample size greater than 20 but less than 100, and the tail category contains categories with a sample size smaller than 20.

### 4.2. Experimental Design and Validation

All experiments in this article are based on Python implementation, version 1.7.1. The server system used in the experiment is Ubuntu 20.04, CUDA version 10.1, and the AdamW optimizer and 300 Epochs are used to train the model. The experiment was trained on a NVIDIA A100 40 G × 8 GPU device. The configuration details of the experimental environment are shown in Table 2.

#### 4.2.1. Experimental Results and Analysis of CIFAR100-LT

In this experiment, the backbone network used by ECVL was ResNet-50, and the experiment was conducted on the long-tailed distribution dataset CIFAR100-LT. The experimental results are shown in Table 3. The enhanced contrastive visual language long-tailed classification algorithm proposed in this paper has an accuracy of 20.5% and 17.2% higher in tail categories than RIDE [46] and TADE [53], and an accuracy of 6.7% and 6.0% higher in all categories compared to RIDE [46] and TADE [53], respectively. The F_1_ values are 14.3% and 11.8% higher than RIDE [46] and TADE [53], respectively. ECVL improves the accuracy difference between majority and minority classes, not only improving the performance of majority classes but also improving the recognition accuracy of minority classes. It also proves that using the semantic features of the original label text as supplementary information for classification is helpful in improving the performance of the model.

#### 4.2.2. Experimental Results and Analysis of ImageNet-LT

In this experiment, the comparative experimental results are shown in Table 4. Compared with the long-tailed image classification algorithm that only uses contrastive learning, the enhanced contrastive visual language proposed in this paper has an accuracy of 29.2% higher in tail classes than PaCo [61], 13.6% higher in all classes than PaCo [61], and a F_1_ value of 14.9% higher than PaCo [61]. The accuracy of CWTA in tail classes is 7.9% higher than that of BALLAD [64], 3.4% higher in all classes, and 11.2% higher in F_1_ values than BALLAD [64]. This not only proves that the proposed enhanced momentum contrastive loss function is more effective than only using contrast loss, but also proves that using text–image pairs for pre-training is helpful for improving model performance.

#### 4.2.3. Experimental Results and Analysis of Places-LT

In this experiment, the ECVL algorithm uses ResNet-50 as the backbone network and conducts experiments on the long-tailed distribution dataset Places-LT. The comparative experimental results are shown in Table 5. Compared with the long-tailed image classification algorithm that only uses contrastive learning, the ECVL long-tailed image classification algorithm has an accuracy of 10.1% higher on tail classes than PaCo [61], an accuracy of 6.0% higher on all classes than PaCo [61], and an F_1_ value of 7.3% higher than PaCo [61]; Compared with the comparative visual language model BALLAD [64], the accuracy on the tail class is improved by 1.3%. The experiment shows that the enhanced momentum contrastive loss function in ECVL is more effective than only using the contrast loss function, and it is helpful to train the model by randomly enhancing the samples with insufficient learning after processing the enhanced momentum contrast loss function.

### 4.3. Experimental Design and Validation

The ECVL long-tailed image classification algorithm proposed in this paper uses the visual characteristics of the image itself and the semantic characteristics of the original label text, the enhanced momentum contrastive loss function and RandAugment to complete the long-tailed classification, and performs well on the public long-tailed dataset. In order to verify the effectiveness of enhanced momentum vs. the loss function and random enhancement in the model, this section conducts ablation experimental analysis on them on different public long-tailed distribution datasets, and the experimental results are shown in Table 6, Table 7 and Table 8.

On CIFAR100-LT, the difference in classification accuracy between most categories and minority categories decreased by 1.8% compared with only using enhanced momentum to compare the loss function and neither using enhanced momentum to compare the loss function nor using random enhancement; With the enhanced momentum contrastive loss function and the random enhancement module, the classification accuracy of most categories and minority categories increased by 2.5% and 3.4%, respectively, than without the random enhancement module. On ImageNet-LT, compared with using only the enhanced momentum contrastive loss function module and neither the enhanced momentum contrastive loss function nor the random enhancement module, the difference between the classification accuracy of most classes and minority classes decreased by 0.7%; compared with the loss function and the random enhancement module with enhanced momentum, the classification accuracy of most categories and minority categories increased by 0.7% and 1.2%, respectively. Through analysis, it is found that although the accuracy of all classes is improved by not using the enhanced momentum contrastive loss function or the random enhancement module, there is still a large difference in the accuracy difference between the majority of categories and the minority in the final fine-tuning process; after adding the enhanced momentum contrastive loss function, the accuracy difference between the majority and minority classes has improved, but in some cases there is degradation (such as Places-LT dataset). The enhanced momentum comparison between the loss function and the random enhancement module can improve the overall accuracy and reduce the accuracy difference between the majority and minority.

## 5. Conclusions

This article first analyzes the advantages and disadvantages of existing long-tailed image classification methods, proposes a long-tailed classification algorithm based on enhanced contrastive visual-language, and then elaborates on the algorithm framework, algorithm design details, algorithm design process, and comparative experimental analysis. In addition, this article conducts comparative experiments and ablation research analysis on three long-tailed datasets: CIFAR100-LT, ImageNet-LT, and Places-LT.

Compared with BALLAD method, ECVL on CIFAR100-LT reduces the difference in classification accuracy between majority and minority classes by 5.7%, and increases F_1_ by 8.5%. Compared with BALLAD, ECVL on ImageNet-LT reduces the difference in classification accuracy between majority and minority classes by 1.7%, and increases F_1_ by 11.2%. Compared with BALLAD, the F_1_ of ECVL on Places-LT has increased by 5.8%. On Places-LT, compared with using only the enhanced momentum contrastive loss function module and neither the enhanced momentum contrastive loss function nor the random enhancement module, the difference in classification accuracy between most classes and minority classes decreased by 1.8%. Compared with the non-random enhancement module, the accuracy rate of minority classification and F_1_ of the enhanced momentum contrastive loss function and random enhancement module increased by 0.7% and 1.3%, respectively. The classification accuracy, difference in accuracy between majority and minority categories, F_1_, and convergence of the model in different quantity categories in the experiment have demonstrated the effectiveness of the algorithm proposed in this paper.

The ECVL method can effectively improve the classification accuracy of tail classes and reduce the difference in classification accuracy between head and tail classes. However, due to the use of pre-training mechanisms, the computational complexity of this method is slightly higher. In the future, improvements can be made in accuracy and complexity to further improve the performance of long-tailed classification models.

## Figures and Tables

**Figure 1 sensors-23-06694-f001:**
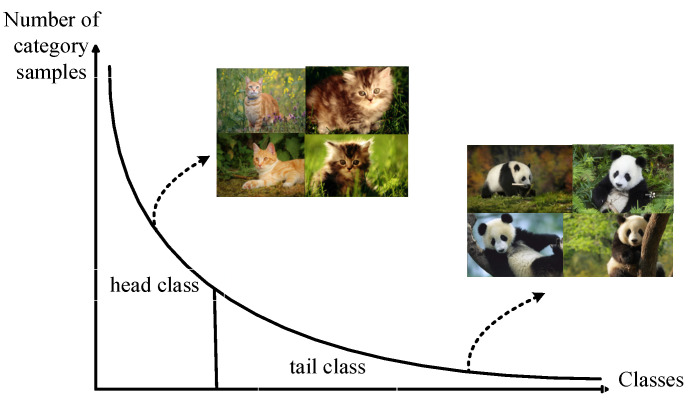
Schematic diagram of the long-tailed distribution of natural animal species.

**Figure 2 sensors-23-06694-f002:**
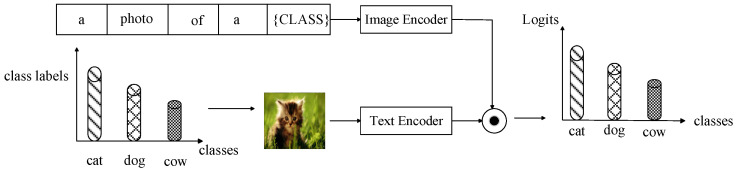
The model processing flow chart of Phase I.

**Figure 3 sensors-23-06694-f003:**
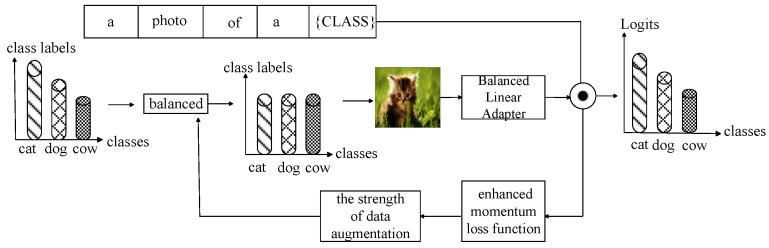
The model processing flow chart of Phase II.

**Table 1 sensors-23-06694-t001:** Long-tailed image classification methods.

Method	Advantages	Limitations
Data Re-Sampling	To some extent, it can reduce the imbalance between head and tail classes.	Causes overfitting of the tail class. Causes underfitting of the head class.
Data Re-Weighting	Assigning weights to different classes and aggravating the learning of tail class.	It is difficult to choose appropriate weights for each class. There may be big differences for different long-tailed datasets.
Data Augmentation	The tail data are extend by data augmentation.	Inability to introduce new effective samples.
Transfer Learning	Transfer the knowledge of head class to tail class.	Requiring a more complex model or module design, which can make the model difficult to train.
Ensemble Learning	Multi-expert model ensembling.	Expert model ensemble requires more computational resources.

**Table 2 sensors-23-06694-t002:** Experimental environment.

Name	Model/Parameter
CPU	Intel(R) Xeon(R) CPU E5-2620 v3 @ 2.40 GHz
GPU	NVIDIA A100 40 G × 8
Memory	128 G
Hard disk	1 T
Operating system	Ubuntu20.04
CUDA	CUDA Version 10.1
Deep learning framework	Pytorch 1.7.1
Development language	Python 3.7

**Table 3 sensors-23-06694-t003:** Experimental results of ECVL on CIFAR100-LT.

Model	Backbone	Accuracy				F_1_
		Head	Medium	Tail	All	
OLTR [63]	ResNet-32	61.8%	41.4%	17.6%	41.2%	52.3%
LDAM [22]	ResNet-32	61.5%	41.7%	20.2%	42.0%	52.9%
cRT [4]	ResNet-32	64.0%	44.8%	18.1%	43.3%	51.9%
RIDE [46]	ResNet-32	69.3%	49.3%	26.0%	49.1%	57.3%
TADE [53]	ResNet-32	65.4%	49.3%	29.3%	49.8%	58.8%
BALLAD [64]	ResNet-50	62.4%	52.3%	38.2%	51.6%	62.1%
ECVL	ResNet-50	65.0%	57.2%	46.5%	55.8%	70.6%

**Table 4 sensors-23-06694-t004:** Experimental results of ECVL on ImageNet-LT.

Model	Backbone	Accuracy				F_1_
		Head	Medium	Tail	All	
OLTR [63]	ResNeXt-50	43.2%	35.1%	18.5%	35.6%	47.6%
cRT [4]	ResNeXt-50	61.8%	46.2%	27.4%	49.6%	53.7%
LWS [4]	ResNeXt-152	62.2%	50.1%	35.8%	52.8%	-
	ResNeXt-50	60.2%	47.2%	30.3%	49.9%	50.6%
ResLT [52]	ResNeXt-152	63.5%	50.4%	34.2%	53.3%	-
	ResNeXt-50	63.0%	50.5%	35.5%	52.9%	55.2%
Balanced Softmax [28]	ResNeXt-101	63.3%	53.3%	40.3%	55.1%	-
	ResNet-50	66.7%	52.9%	33.0%	55.0%	-
PaCo [61]	ResNeXt-50	67.7%	53.8%	34.2%	56.2%	-
	ResNet-50	65.0%	55.7%	38.2%	57.0%	62.3%
BALLAD [64]	ResNeXt-50	67.5%	56.9%	36.7%	58.2%	-
	ResNet-50	71.0%	66.3%	59.5%	67.2%	66.0%
ECVL	ResNet-50	73.2%	69.8%	67.4%	70.6%	77.2%

**Table 5 sensors-23-06694-t005:** Experimental results of ECVL on Places-LT.

Model	Backbone	Accuracy				F_1_
		Head	Medium	Tail	All	
OLTR [63]	ResNet-152	44.7%	37.0%	25.3%	35.9%	46.4%
cRT [4]	ResNet-152	42.0%	37.6%	24.9%	36.7%	45.5%
LWS [4]	ResNet-152	40.6%	39.1%	28.6%	37.6%	46.2%
ResLT [52]	ResNet-152	39.8%	43.6%	31.4%	39.8%	51.2%
PaCo [61]	ResNet-50	37.5%	47.2%	33.9%	41.2%	52.3%
BALLAD [64]	ResNet-50	46.7%	48.0%	42.7%	46.5%	56.8%
	ResNet-101	48.0%	48.6%	46.0%	47.9%	-
	ViT-B/16	49.3%	50.2%	48.4%	49.5%	-
ECVL	ResNet-50	48.6%	48.3%	44.0%	47.2%	59.6%

**Table 6 sensors-23-06694-t006:** Ablation experiment of ECVL on CIFAR100-LT.

Module	Accuracy				F_1_
	Head	Medium	Tail	All	
no momentum contrast loss + no random augment	62.4%	52.3%	38.2%	51.6%	62.1%
momentum contrast loss	62.5%	53.3%	40.1%	52.4%	65.8%
momentum contrast loss + random augment	65.0%	57.2%	46.5%	55.8%	70.6%

**Table 7 sensors-23-06694-t007:** Ablation experiment of ECVL on ImageNet-LT.

Module	Accuracy				F_1_
	Head	Medium	Tail	All	
no momentum contrast loss + no random augment	71.0%	66.3%	59.5%	67.2%	66.0%
momentum contrast loss	72.5%	68.7%	63.2%	69.4%	70.8%
momentum contrast loss + random augment	73.2%	69.9%	67.4%	70.6%	77.2%

**Table 8 sensors-23-06694-t008:** Ablation experiment of ECVL on Places-LT.

Module	Accuracy				F_1_
	Head	Medium	Tail	All	
no momentum contrast loss + no random augment	46.7%	48.0%	42.7%	46.2%	56.8%
momentum contrast loss	47.0%	47.5%	43.2%	46.5%	58.3%
momentum contrast loss + random augment	48.6%	48.3%	44.0%	47.2%	59.6%

## Data Availability

Not applicable.

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
