# Peer review of "A Long-Tailed Image Classification Method Based on Enhanced Contrastive Visual Language"

_sensors, 2023, doi:10.3390/s23156694_

Round 1
Reviewer 1 Report
The paper addresses imbalance of datasets in classification problems (i.e the number of samples in each class are not equilibrated). In this paper, the authors focus on enhanced contrastive visual-language. The proposal is interesting and results are convincing and some ablation studies are performed.
The writing can be improved. The state of the art is very long and not easy to read. It seems sometimes like a list of works. For example, page 3, most sentences have the same structure (starts by In 2020 or In 2023 etc). The authors should better put forward what is related to their work. It is not necessary to describe precisely each paper, but it would be interesting to make some sub-categories.
The semantics and language only appear in the part 3 method, and seems a bit decorrelated from the state of the art. I would have seen the state of the art on visual-language classification methods, then a state of the art which would be more specific on the long-tailed problem (and perhaps more specific to this kind of visual-language techniques)
Equations have to be improved : please adapt the size of the brackets/parentheses depending on the height of the formula
A parenthesis is missing in equation 9
Pages 470-->476 : there is an algorithm in the paragraph
L 575 : replace chapter by paper
There is a problem with the name of the tables (there are 2 tables 5)
There are some sentences to be checked. For example there is no subject in this sentence :"Synthesize samples based on feature dictionaries and enrich the diversity of minority class data by fine-tuning classifiers"
Author Response
Response to Reviewer 1 Comments
Point 1: The writing can be improved. The state of the art is very long and not easy to read. It seems sometimes like a list of works. For example, page 3, most sentences have the same structure (starts by In 2020 or In 2023 etc). The authors should better put forward what is related to their work. It is not necessary to describe precisely each paper, but it would be interesting to make some sub-categories.
Response 1: That’s a good question. The content of the relevant work section has been improved in the paper.
Point 2: The semantics and language only appear in the part 3 method, and seems a bit decorrelated from the state of the art. I would have seen the state of the art on visual-language classification methods, then a state of the art which would be more specific on the long-tailed problem (and perhaps more specific to this kind of visual-language techniques)
Response 2: I'm very sorry for any confusion I may have caused you. The explanation of the long tail image classification method in the third part has been improved.
Point 3: Equations have to be improved : please adapt the size of the brackets/parentheses depending on the height of the formula
A parenthesis is missing in equation 9
Response 3: Thanks for your suggestion. The size of parentheses has been adjusted for the height of the formula in the paper.
Point 4: Pages 470-->476 : there is an algorithm in the paragraph
Response 4: I’m awfullys orry. Algorithm 2 has been modified in the paper.
Point 5: replace chapter by paper
Response 5: This is such a good suggestion that we have replaced this chapter with paper.
Point 6: There is a problem with the name of the table (there are 2 tables 5)
Response 6: Sorry, we changed the name of the table.

Reviewer 2 Report
The manuscript is well written and organized. The argument is original and aligned with the scope of the journal. According to my opinion it can be accepted for publication after improvements.
1. In the introduction, the novelty of the article, with reference to the previous contributions in the literature, needs to be better explained.
2. In the literature background it would be useful to have a table summarizing the limitations of the contributions in the literature. So that we can understand which of them, this study can overcome.
3. When the method is explained, a block diagram would make sense.
4. Advantages and repercussions of the method need to be better explained.
5. In this regard, I suggest to the authors to hypothesize the future application of the method in future technologies. Therefore, the authors could venture some examples from patents and some qualitative evaluations on the improved sustainability, the method can ensure. This is a fundamental aspect to improve the efficacy of the manuscript. For this purpose, please consider the framework of: “Spreafico, C., Landi, D., & Russo, D. (2023). A new method of patent analysis to support prospective life cycle assessment of eco-design solutions. Sustainable Production and Consumption, 38, 241-251”.
6. In the conclusions, the limitations of this study should be better specified.
The English language needs to be revised. There are sentences that are too long. See for example the first sentence of the abstract: “To solve the problem that the common long-tailed classification method does not use the semantic features of the original label text of the image, and the difference between the classification accuracy of most classes and minority classes is large, the long-tailed image classification method based on enhanced contrast visual language trains the head class and tail class samples separately, uses text image to pre-train the information, and uses enhanced momentum contrast loss function and RandAugment enhancement to improve the learning of tail class samples”.
Author Response
Response to Reviewer 2 Comments
Point 1: In the introduction, the novelty of the article, with reference to the previous contributions in the literature, needs to be better explained.
Response 1: Thanks for your suggestion, we have added the contribution of research work in the introduction section.
Point 2: In the literature background it would be useful to have a table summarizing the limitations of the contributions in the literature. So that we can understand which of them, this study can overcome.
Response 2: I'm awfully sorry, we have summarized the advantages and disadvantages of existing long tail image classification methods in relevant work and attached a table.
Point 3: When the method is explained, a block diagram would make sense.
Response 3: I'm very sorry for any confusion I may have caused you. In the method explanation section of the third part, we have attached the structure diagram of the method.
Point 4: Advantages and repercussions of the method need to be better explained.
Response 4: That’s a good question. We have added the advantages and impacts of the proposed method in the third part.
Point 5: In this regard, I suggest to the authors to hypothesize the future application of the method in future technologies. Therefore, the authors could venture some examples from patents and some qualitative evaluations on the improved sustainability, the method can ensure. This is a fundamental aspect to improve the efficacy of the manuscript. For this purpose, please consider the framework of: “Spreafico, C., Landi, D., & Russo, D. (2023). A new method of patent analysis to support prospective life cycle assessment of eco-design solutions. Sustainable Production and Consumption, 38, 241-251”.
Response 5: That’s a very good suggestion. We have supplemented the practical application and contribution of this study in the conclusion section.
Point 6: In the conclusion, the limitations of this study should be better stated.
Response 6: With great apologies, we have elaborated on the contributions and limitations of this study in the conclusion section.

Round 2
Reviewer 1 Report
The paper addresses imbalance of datasets in classification problems (i.e the number of samples in each class are not equilibrated). In this paper, the authors focus on enhanced contrastive visual-language. The proposal is interesting and results are convincing and some ablation studies are performed.
The writing has been improved after first round by better structuring the state of the art section.
Some explanations have been added.
Please be careful about the proofreading of the paper : for example "2.3. Data Augentation"
For me, the equations are not nicely edited (with Word?), hopefully it will be corrected by the edition team. For example, the real domain is not only an upper R ; equations 9, 10 have still brackets that are too small.
Some sections have no introduction : 4.1. Long-tailed Datasets for example
In brief, the writing should be more rigorous.
Author Response
Point 1: Please be careful about the proofreading of the paper : for example "2.3. Data Augentation"
Response 1: That’s a good question. We have modified the title of section 2.3.
Point 2: For me, the equations are not nicely edited (with Word?), hopefully it will be corrected by the edition team. For example, the real domain is not only an upper R ; equations 9, 10 have still brackets that are too small.
Response 2: Thanks for your suggestion. We have modified the representation of R in the paper and modified formulas 9 and 10.
Point 3: Some sections have no introduction : 4.1. Long-tailed Datasets for example
Response 3: Thanks for your suggestion. We have added the content of section 4.1.
Point 4: In brief, the writing should be more rigorous.
Response 4: That’s a good question. We have reviewed the entire paper.
